# IGSF11-Mediated Immune Modulation: Unlocking a Novel Pathway in Emerging Cancer Immunotherapies

**DOI:** 10.3390/cancers17162636

**Published:** 2025-08-13

**Authors:** Sapna Srivastava, Apriliana E. R. Kartikasari, Srinivasa Reddy Telukutla, Magdalena Plebanski, Dibyendu Banerjee

**Affiliations:** 1School of Health and Biomedical Science, Royal Melbourne Institute of Technology (RMIT), Bundoora, VIC 3083, Australia; s4077693@student.rmit.edu.au (S.S.); april.kartikasari@rmit.edu.au (A.E.R.K.); srinivasareddy.telukutla@rmit.edu.au (S.R.T.); 2Department, Academy of Scientific and Innovative Research (AcSIR), Ghaziabad 201002, India; 3Cancer Biology Division, CSIR-Central Drug Research Institute (CSIR-CDRI), Lucknow 226031, India

**Keywords:** IGSF11, cancer, immune checkpoint inhibitors, VISTA

## Abstract

IGSF11 is gaining attention as a promising immune checkpoint molecule that plays a key role in suppressing antitumor immune response. It interacts with the inhibitory receptor VISTA to limit T-cell activity within the tumor microenvironment. Recent studies suggest that blocking this IGSF11-VISTA interaction not only restores immune function but also triggers the release of antitumor cytokines and chemokines, which may help to convert “cold” tumors that don’t respond well to immunotherapy into “hot” ones that are more easily targeted by the immune system. When used in combination with existing therapies such as anti-PD-1/PD-L1, IGSF11-VISTA axis inhibition enhances treatment outcomes by boosting inflammation and immune cell infiltration, making it a powerful immune modulator in emerging cancer therapies.

## 1. Introduction

### 1.1. Overview of Immune Checkpoints in Cancer

Cancer remains one of the biggest global health challenges. It is responsible for nearly one in every six deaths (16.8%) overall, and if we narrow it down to non-communicable diseases (NCDs), that number jumps to nearly one in four (22.8%). Cancer accounts for approximately 30.3% of the early deaths (aged 30 to 69 years) caused by NCDs worldwide. It ranks among the top three causes of death in this age group in the vast majority of countries, 177 out of 183 [1]. Conventional cancer treatments include surgery, chemotherapy, and radiation [2]. Recent advancements in cancer therapies include immunotherapy and targeted therapies, which transform treatment regimens.

Immunotherapy activates immune cells to target tumor cells, with immune checkpoint inhibitors (ICIs) being a major breakthrough. These inhibitors prevent immune tolerance by blocking regulatory pathways such as CTLA-4 and PD-1. The foundation of ICIs began in the early 1980s with James Allison’s discovery of tumor antigens and the T-cell receptor, followed by the identification of CTLA-4 in 1987. Although its function remained unclear for years, Allison’s 1995 research [3] revealed CTLA-4 as a negative regulator of T-cell activation. This led to the development of ipilimumab, the first CTLA-4 blocking antibody, approved by the FDA in 2011 for metastatic melanoma [4].

Programmed Cell Death Protein 1 (PD-1) was later discovered by scientists as an important immune checkpoint inhibitor. Nivolumab was the first PD-1 inhibitor introduced in 2014, marking the beginning of a new era of cancer immunotherapy [5]. Since then, a number of inhibitors targeting PD-1 and its ligand PD-L1, such as pembrolizumab, atezolizumab, durvalumab, and avelumab, have gained FDA approval for use in various malignancies [6]. The year 2019 marked an expansion of the application of atezolizumab to treat triple-negative breast cancer (TNBC) [7]. Although immune checkpoint inhibitors (ICIs) have transformed cancer therapy, limitations such as low response rates with only 20–30% of overall patients benefiting, hence, new predictive biomarkers are urgently needed [8]. Under normal conditions, anti-PD-1/PD-L1 blockade therapy interferes with the binding of PD-1 on T-cells to PD-L1 on cancer cells, resulting in T-cell activation and tumor cell destruction. In some patients, resistance to PD-1/PD-L1 inhibition is linked to the compensatory increase in expression of other immune checkpoint ligands (Figure 1). Researchers are also exploring combinations of ICIs with traditional treatments, including chemotherapy and radiotherapy, to improve therapeutic outcomes. However, ICIs have drawbacks, such as interference with T-cell tolerance to the body’s own antigens and cause immune-related adverse events (irAEs) [9]. Despite these limitations, immunotherapy has continued to evolve rapidly.

### 1.2. The Emergence of Novel Immune Checkpoint Ligands

Cancer immunotherapy has become a cornerstone of modern oncology, offering durable responses through approaches such as immune checkpoint blockade (ICB), adoptive T-cell therapies, and cancer vaccines. While ICB therapies targeting PD-1, PD-L1, and CTLA-4 have revolutionized treatment, their benefits are limited to a subset of patients. Primary resistance remains a major challenge, particularly with PD-1/PD-L1 inhibitors. Although PD-L1 expression is widely used as a predictive biomarker, clinical data show that over 50% of patients with high PD-L1 levels still fail to respond, underscoring the complexity of immune resistance mechanisms beyond PD-L1 expression [10]. This highlights the complexity of immune resistance mechanisms that extend beyond PD-L1 levels.

In addition to primary resistance, acquired resistance is frequently observed in patients who initially respond to the therapy. Over time, these patients may experience disease progression or relapse, often within a few years, underscoring the need for a deeper understanding of resistance mechanisms and the development of alternative or combination therapeutic strategies. Given the significant adverse effects and high financial burden associated with immune checkpoint inhibitors, it is essential to identify additional reliable predictive biomarkers to choose patients more effectively to include those who are most likely to benefit from these treatments.

Multiple immune checkpoints play a crucial role in maintaining self-tolerance by suppressing inappropriate immune responses. Moreover, tumors can exploit all these pathways to escape immune surveillance. Notably, aberrant expression of immune checkpoints, including PD-1 and its ligands (PD-L1/2), CTLA-4, V-domain immunoglobulin suppressor of T-cell activation (VISTA), T-cell immunoglobulin and mucin-domain containing-3 (TIM-3) and lymphocyte-activation gene 3 (LAG-3) has all been detected in a variety of human cancers [11]. Consequently, there is a growing interest in therapies targeting multiple immune checkpoints as a promising strategy to modulate the immunosuppressive tumor microenvironment (TME) and enhance anti-tumor immunity.

Specifically, resistance to PD-1/PD-L1 therapies is increasingly observed, and is often driven by the upregulation of other immune checkpoint molecules and their ligands [12]. This highlights the urgent need to identify alternative immune checkpoints that are overexpressed in tumors. Evidence suggests that blocking a single checkpoint can trigger compensatory mechanisms, leading to upregulation of other checkpoint receptors within the TME. For example, preclinical models have demonstrated that the inhibition of PD-1/PD-L1 can induce LAG-3 expression as a compensatory response. A comparable compensatory response between TIM-3 and PD-1 has been observed in lung cancer [13] and melanoma [14]. According to Du et al. [15], FGL1 a key ligand of the immune checkpoint receptor LAG-3, is predominantly expressed on the surface of breast cancer cells, suggesting its potential role in tumor immune evasion. Similarly, CD155 a ligand for TIGIT, has been implicated in promoting the proliferation and survival of colorectal cancer cells [16]. Research conducted by Kang et al. [17] demonstrated that galectin-9 also triggers apoptosis in TIM3+CD8+T-cells in colon cancer. Furthermore, elevated VSIG-3/IGSF11 (a ligand of VISTA) expression has been observed in colorectal, hepatocellular, and intestinal gastric cancers [18].

Among the emerging immune checkpoint ligands, IGSF11 has been gaining increasing attention due to its potential role in immune evasion [19]. As a binding partner of VISTA, IGSF11 contributes to T-cell suppression within the TME, thereby facilitating tumor immune escape. Recent studies have revealed overexpression of IGSF11 in several malignancies, including colorectal, gastric, hepatocellular [20], and breast cancers [21]. Notably, high levels of IGSF11 have been associated with poor prognosis, reduced infiltration of cytotoxic T-cells, and resistance to immune checkpoint blockade therapy. These findings suggest that IGSF11 could serve as a biomarker of cancer progression and a novel therapeutic target, particularly in tumors exhibiting VISTA-mediated immunosuppression. Given its increasing relevance, a deeper understanding of IGSF11 expression patterns, biological functions, and mechanistic roles in the TME is essential for the effective development of next generation immunotherapeutic strategies.

## 2. Structural and Functional Overview of IGSF11

### 2.1. Gene and Protein Structure

Recently, the V-Set and immunoglobulin (Ig) domain-containing (VSIG) family, which is classified under IgSF and shares structural similarities with the B7 family proteins, has gained recognition as a possible immune checkpoint that plays a role in tumor evasion. Currently, this family consists of eight members, namely VSIG1, VSIG2, VSIG3, VSIG4, VSIG8, VSIG9, VSIG10, and VSIG10 L, all of which are type I transmembrane proteins expressed by various immune and nonimmune cells, many of which exhibit immunosuppressive characteristics [22]. Among them, IGSF11, also known as BT-IgSF, BTIGSF, CT119, CXADRL1, and VSIG3 is a notable member. The protein has a molecular weight of 46 kDa and consists of 431 amino acids, with its genetic location found on chromosome 3q13.32.

Notably, two isoforms of IGSF11 transcripts were predicted: IGSF11-1 and IGSF11-2; however, only the IGSF11-2 transcript was detected, whereas the IGSF11-1 transcript was not observed. The IGSF11-2 protein shows sequence similarity to various members of the immunoglobulin superfamily, demonstrating 37% identity with human CXADR, 34% identity with the Xenopus CTX homolog-like (CTXL), and 30% identity with the endothelial cell-selective adhesion molecule (ESAM). CXADR serves as a receptor on the cell surface for group B coxsackieviruses and the majority of adenoviruses. CTXL, which is the human equivalent of a thymocyte receptor found in Xenopus, is found in several tissues such as the stomach, colon, prostate, trachea, and thyroid gland. The ESAM protein is crucial for cell adhesion and the development of blood vessels. These proteins typically have a structural arrangement that includes a signal peptide, an extracellular domain resembling a V-type immunoglobulin, followed by a C2-type immunoglobulin domain, a transmembrane segment, and a cytoplasmic tail. Similarly, the anticipated structure of IGSF11-2 consists of a hydrophobic signal sequence, two immunoglobulin domains, a transmembrane section, and a cytoplasmic area that features a PDZ-binding domain, suggesting that IGSF11-2 is probably a type I transmembrane protein [19].

The V-type and C-type immunoglobulin-like domains play a key role in VISTA binding. Interestingly, the crystal structure of IGSF11’s extracellular region has been resolved at a 2.64-angstrom resolution, offering valuable insight that could aid in the development of antibodies targeting IGSF11 [23].

### 2.2. Tissue-Specific Expression

IGSF11is primarily reported in a limited number of specific tissues, with the highest expression observed in the testis and brain. Lower levels are also detected in the ovary, adrenal glands, kidney, and thyroid [19].

### 2.3. Known Physiological Roles

IGSF11 plays multiple roles in cellular processes, including cell adhesion, migration [24], proliferation, and differentiation [25]. It is also involved in the formation of synapses [26], maintaining the blood-testis barrier [27], and supporting meiotic processes in both somatic and germ cells [28]. In addition to these functions, IGSF11 acts as a ligand for the immune checkpoint protein VISTA, helping to regulate immune cell activity, particularly in T-cells (Figure 2). In zebrafish, it is essential to establish adult pigment patterns by influencing melanophore migration and survival [26].

IGSF11 has also been recognized as a ligand for VISTA, which is also referred to as PD-1H [29], leading to an inhibitory effect on T-cell activity [30]. Watanabe and colleagues [20] further reported that IGSF11 expression is upregulated in colorectal cancer, hepatocellular carcinoma, and intestinal-type gastric cancer. Ghouzlani and colleagues [31] indicate that the interaction between IGSF11 and VISTA on activated T-cells leads to the suppression of T-cell proliferation and a decrease in cytokine and chemokine production. Shekari and colleagues [19] showed that decreasing the interactions between IGSF11 and VISTA after silencing IGSF11 could change the cytokine profile of human T-cells towards a more pro-inflammatory state, which aids T-cells in becoming more activated and transforming the immunosuppressive tumor microenvironment into a more anti-tumoral one.

## 3. IGSF11 in Cancer: Expression Patterns and Clinical Correlations

### 3.1. Differential Expression Across Cancer Types

While IGSF11 shows minimal expression in normal tissues, recent studies show that it is significantly upregulated in various cancers, such as colorectal cancer, hepatocellular carcinoma, melanoma [19], gastric cancer [20], and breast cancer [21]. In breast cancer models, IGSF11 expression appears to be influenced by TGF-β signaling. TGF-β activates EMT-related pathways, which in turn promote the expression of the long non-coding RNA Platr18, ultimately leading to increased IGSF11 levels. This regulatory mechanism may influence the metastatic behavior of breast cancer cells [23]. The following sections explore the role of IGSF11 in specific cancer types in detail, and highlight its biological significance and clinical implications (Table 1).

#### 3.1.1. Glioblastoma

Gliomas are the most common and aggressive type of brain tumors in adults, and contribute significantly to brain tumor–related mortality [32]. Despite the use of conventional therapies for glioblastoma, patient prognosis remains poor [33]. Although immunotherapy has revolutionized cancer treatment, most glioma patients show a limited response to the blockade of traditional immune checkpoint pathways [34].

Ghouzlani and colleagues [31] have shown that IGSF11 expression was markedly elevated in patients with high-grade gliomas. In addition, IGSF11 protein has been detected in various glioma samples, supporting its relevance at both the transcript and protein levels. Analysis of an independent cohort from The Cancer Genome Atlas (TCGA) further validated the upregulation of IGSF11 transcripts in high-grade gliomas. Moreover, IGSF11 expression was positively correlated with the expression of several critical immune checkpoint molecules, suggesting its involvement in modulating immune responses within the tumor microenvironment. Notably, glioma patients with high IGSF11 levels exhibit increased immune cell infiltration; however, their microenvironment was predominantly immunosuppressive. Importantly, elevated IGSF11 expression was associated with poorer overall survival, indicating its potential as a prognostic biomarker and a therapeutic target in gliomas.

#### 3.1.2. Gastric Cancer

Gastric cancer ranks among the most prevalent cancers worldwide [35]. Improvements in endoscopic diagnostic techniques have facilitated the early detection of gastric tumors, significantly enhancing cure rates through surgical intervention. However, the outlook for patients with advanced gastric cancer remains poor, as current treatment options such as chemotherapy offer limited success. Consequently, the overall 5-year survival rate for advanced-stage disease remains between 5% and 15% [20].

Watanabe and colleagues [20] recognized IGSF11 as a new therapeutic target for gastric cancer and found two peptide sequences originating from IGSF11 that could provoke cytotoxic T lymphocyte (CTL) responses directed at IGSF11-expressing gastric cancer cells in an HLA-A*0201-restricted manner. These findings provide valuable insights into the mechanisms of human carcinogenesis and hold promising potential for the development of clinical immunotherapy strategies.

Targeting the extracellular domain of IGSF11 with an antibody may also offer a promising alternative approach for gastric cancer treatment. Additionally, given that high IGSF11 expression is crucial for cancer cell growth, small-molecule inhibitors that block IGSF11 or its downstream signaling pathways may also serve as promising therapeutic options.

#### 3.1.3. Melanoma

Melanoma, while rare among skin cancers, is notably aggressive and accounts for around 75% of skin cancer-related fatalities. Its prevalence has been on the rise in recent years [36]. Patients with early-stage melanoma generally respond well to treatment, whereas those with advanced-stage disease often have poor response rates and unfavorable prognoses.

The survival rate for people diagnosed with advanced-stage melanoma can drop to as low as 30% [37]. As a highly immunogenic cancer, melanoma progression is significantly influenced by immune-related mechanisms, especially the increased expression of inhibitory immune checkpoints and their ligands on both tumor and immune cells, which promotes tumor immune escape [38]. Therefore, melanoma is widely regarded as a valuable model for assessing the effectiveness of immunotherapies, particularly ICIs [39].

Shekari et al. [19] suggested that although IGSF11 does not serve as a prognostic marker for patients with skin cutaneous melanoma (SKCM), it plays a significant role in promoting the progression of A2058 melanoma cells, indicating its potential as a therapeutic target. Silencing of IGSF11 disrupts its interaction with VISTA, leading to a shift in the cytokine milieu of human T-cells toward a pro-inflammatory profile. This shift enhances T-cell activation and contributes to the reprogramming of the TME from an immunosuppressive to an immunoactive anti-tumoral state. Consequently, targeting IGSF11 boosts T-cell-mediated anti-tumor immune responses and also suppresses the proliferation of A2058 melanoma cells, highlighting its dual therapeutic potential in melanoma treatment.

#### 3.1.4. Breast Cancer

Breast cancer ranks among the most commonly diagnosed cancers across the globe [40]. It is a major contributor to cancer-associated morbidity and mortality in the female population.

Olbromski et al. [21] showed that VISTA/IGSF11/PSGL-1 proteins are present on breast cancer cells as well as in immune cells that infiltrate the tumor, including lymphocytes (CD45+) and macrophages (CD68+). Their observations suggest that increased aggressiveness in breast cancer cells is linked to higher levels of proteins within the VISTA/IGSF11/PSGL-1 axis. These receptors are overexpressed in the most aggressive breast cancer subtypes, as well as on immune cells within the TME.

#### 3.1.5. Pan-Cancer Expression Analysis of IGSF11 and VISTA Using UALCAN

The UALCAN pan-cancer analysis graph [41] (Figure 3) illustrates the differential mRNA expression levels of IGSF11 and VISTA (C10orf54) across various tumor types in comparison to their corresponding normal tissues using the TCGA database. For each gene, the graph displays boxplots representing expression values (typically in transcripts per million) for both tumor (shown in red) and normal samples (shown in blue).

In the case of IGSF11, the UALCAN data revealed significantly elevated expression in multiple cancer types, including glioblastoma multiforme (GBM), skin cutaneous melanoma (SKCM) and lung squamous carcinoma (LUSC) compared with their respective normal tissue counterparts. This overexpression suggests a potential role of IGSF11 in tumor progression and immune modulation (Figure 3a).

Similarly, VISTA (C10orf54) exhibits notable upregulation in several malignancies, with high expression observed in sarcoma (SARC), cholangiocarcinoma (CHOL), and glioblastoma multiforme (GBM). The consistent overexpression of VISTA in tumor tissues relative to normal supports its emerging role as a negative immune checkpoint regulator involved in tumor immune evasion (Figure 3b).

Overall, the UALCAN graph underscores the tumor-specific upregulation of both IGSF11 and VISTA across a range of cancer types compared to normal counterparts (Table 2), reinforcing their relevance as potential targets in immune checkpoint blockade strategies.

On the diagnostic front, the restricted expression pattern of IGSF11 in normal tissues is mainly limited to the testis and brain, enhancing its specificity as a tumor-associated antigen. Immunohistochemical (IHC) detection of IGSF11 protein levels in biopsy samples may aid in stratifying patients based on immune checkpoint activity, particularly in VISTA-dominant cancers where PD-1 expression is low. Furthermore, liquid biopsy approaches to detect IGSF11 mRNA or circulating tumor-derived exosomes carrying IGSF11 may offer minimally invasive diagnostic tools for early detection or monitoring of treatment response.

Given its dual role in immune regulation and tumor progression, IGSF11 holds promise as a predictive biomarker for checkpoint inhibitor responsiveness, as well as a prognostic marker for cancer aggressiveness, and potentially also for immune landscape characterization. However, validation in large, multi-center clinical cohorts and standardized assays will be essential before clinical utility can be confirmed.

### 3.2. Co-Expression with Other Immune Checkpoints

Co-expression of VISTA with other checkpoints, such as PD-1 or TIGIT, can contribute to a more profound state of T-cell dysfunction, allowing tumors to escape immune surveillance. This overlapping expression pattern can also exist in regulatory T-cells (Tregs) and myeloid-derived suppressor cells (MDSCs), further promoting immunosuppression. Recent studies suggest that tumors expressing high levels of co-inhibitory ligands such as PD-L1 and IGSF11 may respond poorly to monotherapies targeting only one pathway [42]. Hence, understanding co-expression profiles is crucial for designing combination immunotherapies that simultaneously target multiple checkpoints.

## 4. Mechanisms of IGSF11-Mediated Immune Modulation

### 4.1. IGSF11-VISTA Signaling Axis

IGSF11 is a specific ligand for VISTA [18]. Co-immunoprecipitation (Co-IP) demonstrated the association between IGSF11 and VISTA, while surface plasmon resonance (SPR) and fluorescence-activated cell sorting (FACS) assays provided additional validation of the specificity of this interaction [29]. These studies also demonstrates that the binding of IGSF11 to VISTA, specifically within the V-type and C-type immunoglobulin-like domains, leads to the suppression of T-cell proliferation and cytokine production. This includes the downregulation of key cytokines such as interleukin-17 (IL-17), chemokine ligand 3 (CCL3), C-X-C motif chemokine 11 (CXCL11), and chemokine ligand 5 (CCL5) [18], providing a theoretical basis for targeting IGSF11 in cancer immunotherapy (Figure 4).

Both IGSF11 and VISTA antibodies have been shown to effectively block these interactions [43]. SG7, an antibody targeting VISTA, exerts its effects by attaching to four crucial epitopes H122, K38, E125, and F36 which coincide with the binding sites of two additional VISTA antibodies, BMS767 and VSTB112 (created by Bristol Myers Squibb). SG7 is capable of competing with both antibodies and restoring T-cell function [44].

The interaction between IGSF11 and VISTA was influenced by the acidic TME. At physiological pH (7.4), the binding affinity is approximately 20 nM, whereas at pH 6.0, binding affinity is commonly observed in the TME is 80 nM [44].

HMBD-002 is another therapeutic antibody that targets the CC’ loop of VISTA. It effectively disrupts the IGSF11–VISTA interaction and promotes interferon-gamma (IFN-γ) production from IGSF11-stimulated T-cells. The affinity of HMBD-002 for VISTA is also pH-dependent, with optimal binding observed in the pH range of 5.5 to 7.5 [45].

### 4.2. Role in T-Cell Suppression and Immune Evasion

Both IGSF11 and VISTA play inhibitory roles in immune regulation. They are known to suppress the activity of tumor-infiltrating lymphocytes (TILs), particularly CD8+ T-cells. Furthermore, both molecules enhance the release of immunosuppressive cytokines such as TGF-β and IL-10, while decreasing the generation of pro-inflammatory cytokines such as TNF-α and IFN-γ, which ultimately aids in creating an immunosuppressive tumor milieu [19].

To date, the immune regulatory role of IGSF11 has been demonstrated in a small number of cancer types, whereas VISTA’s involvement has been validated in a broader range of tumors. This provides a significant opportunity for further research into IGSF11’s role in tumor immune regulation, especially since its intracellular mechanisms remain largely unknown.

## 5. Preclinical and Clinical Insights into IGSF11 Targeting

### 5.1. Potential Therapeutic Strategies: Antibodies and Small Molecule Inhibitors

Monoclonal antibodies designed to block IGSF11 may disrupt its interaction with VISTA, and restore T-cell activity within the tumor microenvironment. Furthermore, the development of small molecule inhibitors that interfere with IGSF11’s intracellular signaling. These strategies may provide new avenues for overcoming resistance to current immunotherapies. Monoclonal antibodies such as IMT-18 disrupt the IGSF11–VISTA interaction, restoring T-cell function within the tumor microenvironment. HMBD-002 binds to a critical domain of VISTA involved in IGSF11 engagement, while SG7 directly blocks the VISTA–IGSF11 interface. Additionally, small molecule inhibitors like K284-3046 and sinefungin are being explored to interfere with IGSF11’s signaling pathways, further enhancing immune activation against tumors (Table 3).

#### 5.1.1. IMT-18

iOmx Therapeutics has developed next-generation cancer immunotherapies with a focus on IMT-18, an antibody targeting IGSF11 to overcome tumor-induced immune suppression, particularly in PD-1/PD-L1-resistant tumors. Using their proprietary iOTarg™ platform, iOmx aimed to discover and develop novel immune checkpoint modulators exploited by cancer cells to evade immune responses. Targeting IGSF11 with IMT-18 represents a promising strategy to enhance the efficacy of cancer immunotherapy [46].

#### 5.1.2. HMBD-002

HMBD-002 is a first-in-class IgG4 monoclonal antibody developed by Hummingbird Biosciences that specifically targets VISTA, a key immunoregulatory protein interacting with IGSF11 and leucine-rich repeats and immunoglobulin-like domains 1 (LRIG1) [19]. HMBD-002 primarily binds to the C–C’ loop of VISTA, a critical domain involved in ligand interactions, effectively neutralizing VISTA’s immunosuppressive function without depleting VISTA-expressing cells, due to its Fc-independent mechanism of action [44]. Functionally, HMBD-002 disrupts the suppressive VISTA–IGSF11 signaling axis, enabling anti-CD3-stimulated T-cells to produce IFN-γ and reversing the inhibitory influence of myeloid-derived suppressor cells (MDSCs) on T-cell responses. It also suppresses tumor cell invasion and promotes T-cell polarization toward Th1/Th17 phenotypes. In multiple preclinical models, including humanized and syngeneic murine models of breast, colorectal, and lung cancers, HMBD-002 demonstrated potent anti-tumor effects with minimal toxicity. Notably, when combined with pembrolizumab, enhanced efficacy was observed, particularly in tumors enriched with MDSCs. These encouraging outcomes have resulted in its continued assessment in a Phase 1/2 clinical trial (NCT05082610), both as a monotherapy and in combination with pembrolizumab for individuals with advanced solid tumors exhibiting elevated VISTA expression [47]. Collectively, these findings underscore the therapeutic relevance of targeting the VISTA–IGSF11 axis in immunooncology.

#### 5.1.3. SG7

SG7 is a cross-reactive monoclonal antibody developed via yeast surface display that binds VISTA with high affinity across human, murine, and cynomolgus monkey species. It blocks VISTA’s interactions with IGSF11 at a physiological pH of 7.4 and with PSGL-1 at an acidic pH of 6.0, primarily by affecting the histidine 122 and glutamic acid 125 residues. Functionally, SG7 restores T-cell activation in Jurkat assays, reduces tumor growth in syngeneic mouse models, including the 4T1 breast cancer model, and remodels the tumor microenvironment by lowering polymorphonuclear myeloid-derived suppressor cells (PMN-MDSCs) and increasing CD4+ and CD8+ T-cells, without affecting other myeloid populations. In a melanoma mouse model, the administration of SG7 at 10 mg/kg for two weeks effectively slowed tumor progression. Additionally, combining SG7 with anti-PD-1 therapy in the MC38 colon cancer model produced similar positive results, highlighting the therapeutic potential of targeting the IGSF11–VISTA pathway. Notably, SG7 combined with anti-PD1 antibodies shows superior efficacy over monotherapies in colon adenocarcinoma models [44].

#### 5.1.4. K284-3046

According to Xie et al. [43] protein-ligand docking study of VISTA/IGSF11 led to the identification of K284-3046 as a small molecule inhibitor of IGSF11, and its biological activities were evaluated in vitro. It binds to specific amino acids on IGSF11, namely PRO46, SER48, GLY133, THR134, GLN136, VAL152, GLY154, and THR156. K284-3046 can reverse the inhibitory effect of IGSF11 on activated PBMCs and promote PBMC proliferation. It increases the levels of cytokines such as IFN-γ and IL-17, which are typically reduced by IGSF11. By inhibiting IGSF11 function, K284-3046 may enhance T-cell activity and potentially improve anti-tumor immune responses.

#### 5.1.5. Sinefungine (SFG)

Inhibition of METTL3 can decrease adenosine methylation of miR-125a-5p, leading to reduced IGSF11 expression. Sinefungin (SFG), a METTL3 inhibitor, has been shown to decrease m6A-miR-125a-5p levels, enhance miR-125a-5p association with GW182, and subsequently suppress IGSF11 expression [48].

### 5.2. Challenges and Opportunities in Drug Development

The development of therapeutics that target IGSF11 presents significant challenges and promising opportunities. One major challenge lies in the limited understanding of its precise role in the tumor microenvironment and its pathway of interaction with immune checkpoint molecules such as VISTA. The lack of well-characterized antibodies or small-molecule inhibitors further complicates efforts to therapeutically target this pathway. Additionally, the expression of IGSF11 in certain normal tissues raises concerns about potential off-target effects and immune-related toxicities. Despite these challenges, IGSF11 offers a compelling opportunity as a novel immune checkpoint molecule, particularly in cancers where VISTA-mediated immune evasion is prominent. Advances in structural biology, high-throughput screening, and bioinformatics could accelerate the identification of potent and selective inhibitors. Furthermore, combinatorial strategies targeting IGSF11 along witother checkpoints may enhance anti-tumor immunity and overcome resistance to existing immunotherapies.

## 6. IGSF11 in the Tumor Microenvironment (TME)

### 6.1. IGSF11-VISTA Interaction Is a Driver of Cold Tumor Phenotype

IGSF11 serves as a key modulator of cellular interactions within the TME, orchestrating complex signaling between tumor cells, immune infiltrates, and stromal components. Through its binding to VISTA, a known negative regulator of T-cell activity, IGSF11 transmits immunosuppressive signals that dampen T-cell receptor (TCR) signaling, reduce cytokine secretion, and impair T-cell proliferation. This interaction not only supports tumor immune evasion but also promotes the development of a cold tumoriogenic environment characterized by the enrichment of exhausted CD8^+^ T-cells, regulatory T-cells (Tregs), and myeloid-derived suppressor cells (MDSCs).

VISTA binds to IGSF11 and suppresses T-cell function by decreasing the production of key pro-inflammatory chemokines, including CCL3, CCL5, and CXCL11, as well as cytokines such as IL-2, IFN-γ, and IL-17 in activated T-cells. Moreover, T-cell proliferation was markedly reduced in the presence of IGSF11, highlighting its potent immunosuppressive role. Emerging evidence suggests that VISTA, in addition to its surface expression, accumulates within the cytoplasm of various tumor cells, including breast and non-small cell lung cancer (NSCLC), where it may interact with IGSF11 to initiate intracellular signaling pathways that support tumor cell proliferation. While this proposed non-immune, tumor-intrinsic function of the VISTA–IGSF11 axis is intriguing, it remains a hypothesis that requires further experimental validation [23] (Figure 4).

In a subsequent study, Xie et al. [43] confirmed the direct connection between IGSF11 and VISTA by studying its crystal structure. They showed that the co-culture of activated peripheral blood mononuclear cells (PBMCs) and CD4^+^ T-cells with Chinese hamster ovary (CHO) cells expressing the extracellular domain (ECD) of IGSF11 led to a significant reduction in TNF-α, IFN-γ, IL-17 levels, and T-cell proliferation.

In line with these findings, Ghouzlani et al. [31] reported a correlation between increased IGSF11 expression and elevated TGF-β levels in glioma patients, further suggesting its contribution to immunosuppressive signaling within the TME.

Complementary results were observed in a study conducted by Shekari et al., wherein T-cells that were co-cultured with IGSF11 siRNA-transfected A2058 cells exhibited elevated levels of IFN-γ and IL-12, supporting the notion that disruption of the IGSF11-VISTA axis enhances pro-inflammatory responses and alleviates T-cell suppression [19].

Interestingly, while TNF-α was initially characterized for its anti-tumoral effects, recent studies have revealed its dual role in cancer: TNF-α can promote the expansion and suppressive function of regulatory T-cells (Tregs) [49] and myeloid-derived suppressor cells (MDSCs) [50], thereby contributing to immune suppression. Additionally, TNF-α triggers activation-induced cell death (AICD) in CD8^+^ T-cells and hinders their movement into the TME [51]. Donia et al. [52] further showed that TNF-α generated by CD4^+^ TILs in melanoma inhibits anti-tumor responses, highlighting the intricate and situation-dependent functions of cytokines within the TME.

Collectively, these studies highlight the emerging role of IGSF11 as a critical immunoregulatory molecule within the tumor microenvironment, primarily through its interaction with VISTA and its capacity to modulate cytokine signaling pathways. Therapeutically, targeting the IGSF11 axis may provide a novel means to reverse immune suppression, enhance T-cell-mediated anti-tumor responses, and potentially overcome resistance to existing checkpoint blockade therapies.

### 6.2. Impact on Tumor Progression and Metastasis

IGSF11 is increasingly recognized for its immunosuppressive function as well as its potential role in promoting tumor progression and metastasis. Its interaction with VISTA facilitates immune escape by inhibiting T-cell activation, thereby allowing tumor cells to proliferate and survive in an immune-privileged environment. Beyond immune modulation, emerging evidence suggests that IGSF11 may directly influence tumor cell behavior. High expression levels of IGSF11 have been correlated with aggressive clinical features in several cancers, including gliomas, hepatocellular carcinoma, melanoma, and gastric cancer. Although direct mechanistic studies on IGSF11’s role in metastasis are limited, its dual involvement in tumor immune evasion and microenvironment remodeling strongly suggests that it contributes to both local tumor progression and distant metastasis. Targeting IGSF11 may therefore represent a strategic approach to suppress tumor growth and prevent metastatic spread by restoring immune control and disrupting pro-tumorigenic signaling networks.

### 6.3. Implications for Combination Immunotherapies

The recognition of IGSF11 as an important ligand for the VISTA inhibitory immune checkpoint receptor has revealed the possibilities of novel combination immunotherapy approaches, especially in tumors refractory to existing checkpoint blockade treatments. As opposed to PD-1/PD-L1 or CTLA-4 pathways, the IGSF11-VISTA pathway is capable of maintaining the suppression of the immunity in non-redundant manners, thus representing an influential target for dual- or multi-checkpoint blockade. A combination of IGSF11 inhibition with anti-PD-1 or anti-CTLA-4 treatments can augment the activation of T-cell populations, activate the exhausted T-cell pools, and enhance the repertoire of immunity toward the tumor antigens. It is established in the preclinical models that the disruption of the interaction of IGSF11-VISTA re-establishes the production of IL-2, IFN-γ, TNF-α, and other important cytokines, augmenting anti-tumor immunity and enhancing the proliferation of the T-cell population, which can cooperate with the PD-1/PD-L1 blockade.

Integrated anti-IGSF11 therapy in conjunction with conventional anti-tumor therapy modalities has potential in reversing suppression of the immune system and enhancing therapeutic efficacy. Blocking IGSF11 disrupts its interaction with VISTA, releasing T-cell inhibition and restoring effector functions in the tumor microenvironment. When combined with immunogenic checkpoint inhibitors (like anti-PD-1, anti-PD-L1, or anti-CTLA-4), the blockade would potentially synergize with T-cell activation, proliferation, and cancer cell killing. Additionally, application of the combination of IGSF11 inhibition and chemotherapy/radiotherapy could optimize immunogenic cell death, augmenting exposure of neoantigens and immune recognition. Lastly, application of anti-IGSF11 in combination with CAR-T cell therapy could optimize T-cell infiltration and persistence in solid tumors, correcting significant bottlenecks to the efficacy of CAR-T cell-based therapy. Overall, such combination regimens would greatly optimize tumors clearance, especially in cancers refractory to single-agent monotherapies (Figure 5).

## 7. Future Perspectives and Research Directions

### 7.1. Gaps in Current Understanding

IGSF11 has been recognized as a potential ligand for VISTA, and studies have shown that disruption of the VISTA/IGSF11 interaction can lead to T-cell activation [18]. However, its role in cancer progression remains unclear. To date, there is no available evidence on how silencing IGSF11 in cancer cells might influence human T-cell responses [19].

Moreover, while great efforts have been made to unravel the signaling pathways and epigenetic regulations of VISTA expression, comparatively less attention has been given to IGSF11. Unraveling the regulatory mechanism of IGSF11 expression may disclose new ways of suppressing this molecule and reveal additional therapeutic targets for enhancing the potency of cancer immunotherapy.

### 7.2. Clinical Translation: From Bench to Bedside

Preclinical studies have demonstrated that IGSF11 expression correlates with poor prognosis and immunosuppression across multiple cancers, including breast, gastric, and colorectal malignancies. Functional inhibition of IGSF11, through genetic knockdown or potential antibody-based strategies, has shown enhanced anti-tumor immunity in experimental models, suggesting its therapeutic relevance. While clinical trials directly targeting IGSF11 are currently lacking, its expression profile and interaction with established immune checkpoints offer a strong rationale for translational development. Future efforts should focus on the design and validation of IGSF11-specific inhibitors and integration into combination immunotherapy regimens, particularly for patients unresponsive to PD-1/PD-L1 blockade.

## 8. Conclusions

IGSF11 has emerged as a critical immune regulatory molecule with significant implications in cancer biology. As a ligand for the immune checkpoint receptor VISTA, IGSF11 plays a central role in suppressing T-cell activation and facilitating immune evasion in the tumor microenvironment. Its overexpression has been documented in several cancer types, and correlates with poor prognosis and aggressive tumor behavior. Functional studies have demonstrated that genetic silencing or inhibition of IGSF11 enhances anti-tumor immunity. Importantly, targeting IGSF11 enhances prostimulatory cytokine production, promotes T-cell recruitment, and reinvigorates anti-tumor immune responses, effectively converting immunologically “cold” tumors into “hot” ones. Moreover, combining IGSF11 inhibition with existing immunotherapies, such as anti PD-1/PD-L1 antibodies, demonstrates synergistic effects that, further amplify immune activation and therapeutic efficacy. Some inhibitors that target the VISTA-IGSF11 axis, like SG7 show improved efficacy in combination with anti-PD-1 therapy in colon cancer models. Another inhibitorHMBD-002, is undergoing phase 1/2 clinical trials as monotherapy and in combination with pembrolizumab. These insights position IGSF11 as a promising therapeutic target to overcome immunotherapy resistance and improve clinical outcomes. Nonetheless, translating these findings into clinical practice remains a significant challenge, underscoring the need for continued research into the molecular mechanisms and therapeutic potential of IGSF11 in cancer immunotherapy.

## Figures and Tables

**Figure 1 cancers-17-02636-f001:**
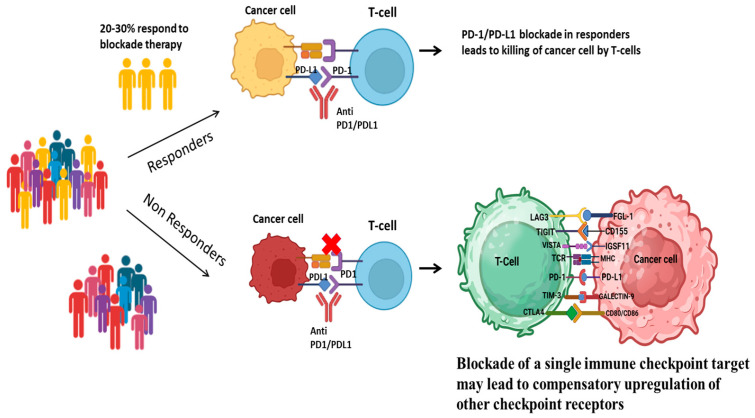
Potential mechanism of resistance to PD-1/PD-L1 blockade therapy in cancer cells.

**Figure 2 cancers-17-02636-f002:**
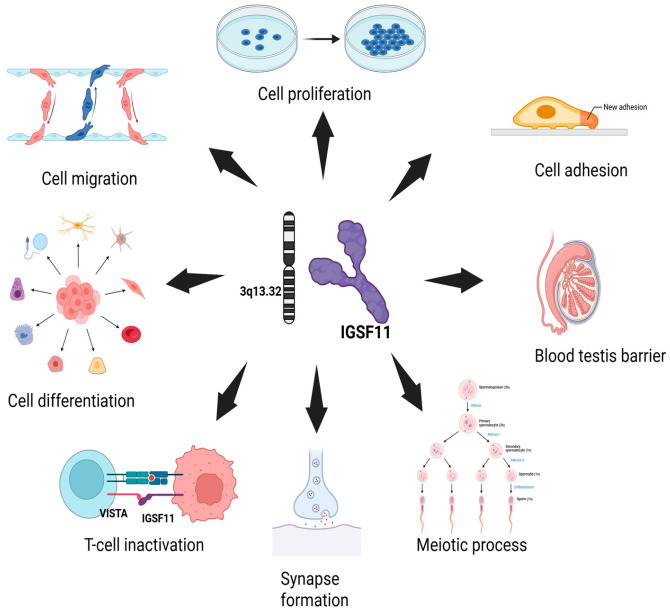
Functional role of IGSF11 in various cellular processes.

**Figure 3 cancers-17-02636-f003:**
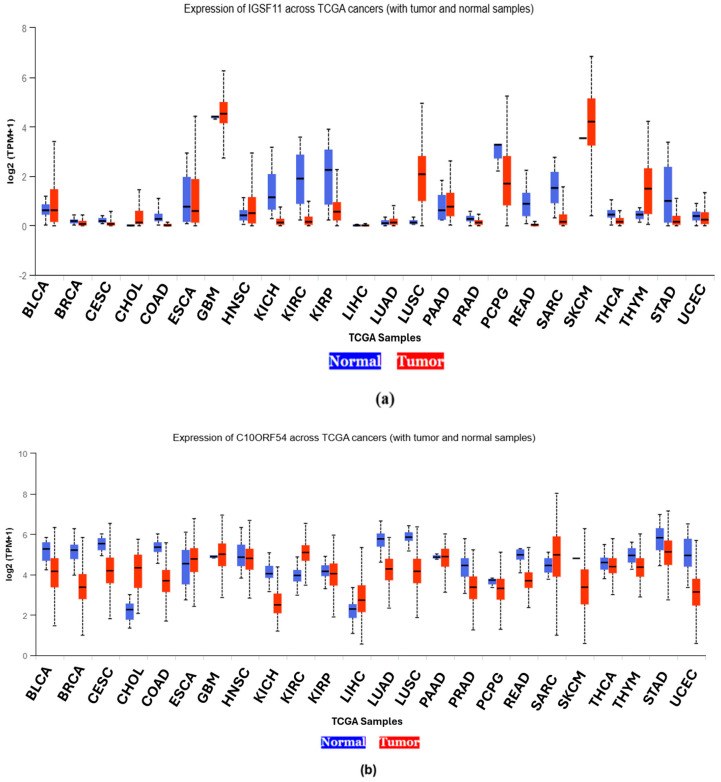
Pancancer expression of IGSF11 and VISTA using UALCAN. (**a**) IGSF11 and (**b**) VISTA (C10orf54) mRNA expression levels across different tumor types based on TCGA RNAseq data analysed through the UALCAN portal. Expression is presented as log2 (TPM + 1).

**Figure 4 cancers-17-02636-f004:**
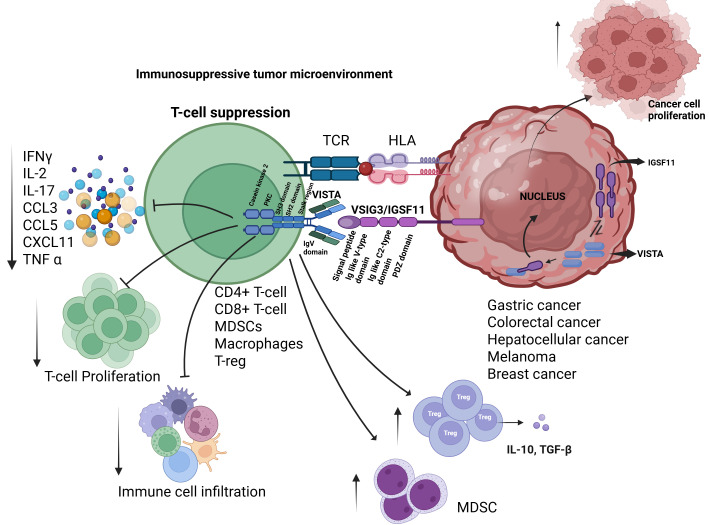
IGSF11–VISTA interaction contributes to an immunosuppressive tumor microenvironment. Binding of IGSF11 and VISTA inhibits prostimulatory cytokine production, T-cell activation, and immune cell infiltration, promoting immune evasion. Possibly, VISTA and IGSF11 are localized in the cytoplasm of some cancer cells, including breast and Non-small cell lung cancer, where their binding may also regulate tumor cell proliferation.

**Figure 5 cancers-17-02636-f005:**
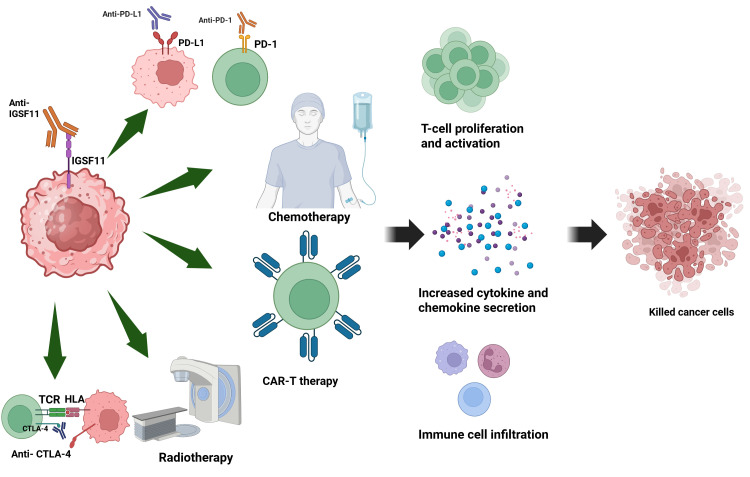
Cancer therapy combining anti-IGSF11 targeting immune checkpoint inhibitors, chemotherapy, radiotherapy, or CAR-T cells. Blocking IGSF11 relieves immune suppression, and when combined with other therapies, enhances T-cell activation, infiltration, and tumor cell killing.

**Table 1 cancers-17-02636-t001:** IGSF11 expression in different cancer types.

Cancer Type	Cell Lines or Patient Samples	Reference
Gastrointestinal cancer	Human gastric cell line MKN1, MKN28 and MKN45, MKN74, Kato III St-4 and human colon cancer cell line SNU-C4	[19]
Hepatocellular cancer	Human hepatocellular carcinoma cell line SNU475	[19]
Glioblastoma	Twenty glioma patients and thirty-two PBMC specimens	[30]
Melanoma	Human melanoma cell line A2058	[18]
Breast cancer	Human breast cancer cell line Me16C, T47D, MCF7,BT474 SKBR3, MDAMB231, MDAMB231/BO2	[20]

**Table 2 cancers-17-02636-t002:** Expression levels of IGSF11 and VISTA in various cancers based on publicly available transcriptomic data at UALCAN compared to their normal counterparts.

Abbreviation	Cancer Type	IGSF11 Expression	VISTA Expression
BLCA	Bladder urothelial carcinoma	High	High
BRCA	Breast invasive carcinoma	Low	Low
CESC	Cervical squamous cell carcinoma	Low	High
CHOL	Cholangiocarcinoma	High	High
COAD	Colon adenocarcinoma	Low	Low
ESCA	Esophageal carcinoma	High	High
GBM	Glioblastoma multiforme	High	High
HNSC	Head and Neck squamous cell carcinoma	High	High
KICH	Kidney Chromophobe	Low	Low
KIRC	Kidney renal clear cell carcinoma	Low	Low
KIRP	Kidney renal papillary cell carcinoma	Low	High
LIHC	Liver hepatocellular carcinoma	Low	Low
LUAD	Lung adenocarcinoma	High	Low
LUSC	Lung squamous cell carcinoma	High	Low
PAAD	Pancreatic carcinoma	High	High
PRAD	Prostate adenocarcinoma	Low	Low
PCPG	Pheochromocytoma and paraganglioma	High	High
READ	Rectal adenocarcinoma	Low	Low
SARC	Sarcoma	Low	High
SKCM	Skin cutaneous melanoma	High	High
THCA	Thyroid carcinoma	Low	High
THYM	Thymoma	High	High
STAD	Stomach adenocarcinoma	Low	High
UCEC	Uterine Corpus Endometrial Carcinoma	High	Low

**Table 3 cancers-17-02636-t003:** Status of IGSF11 targeting mAbs and small molecule as next generation therapeutics.

Drug	Description	Phase Target	Target	Cancer Type	Status	Developed by
IMT-18	mAb	Preclinical	IGSF11	Targeting tumors that are resistant to PD-1/PD-L1 therapies	-	IOMx [45]
HMBD-002	mAb	Phase I/II NCT05082610	VISTA/IGSF11	Advanced solid tumors	Active, Not Recruiting	Hummingbird Bioscience [46]
SG7	mAb	Preclinical	VISTA/IGSF11 and PSGL1	Colon cancer, melanoma	-	Mehta et al. 2021 [43]
K284-3046	Small molecule	Preclinical	IGSF11	In vitro assays	-	Xie et al. 2021 [42]
Sinefungin	Small molecule	Preclinical	IGSF11	In vitro assays	-	Bougras-Cartron G et al. 2023 [47]

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
