# Peer review of "IGSF11-Mediated Immune Modulation: Unlocking a Novel Pathway in Emerging Cancer Immunotherapies"

_cancers, 2025, doi:10.3390/cancers17162636_

Round 1
Reviewer 1 Report
Comments and Suggestions for Authors
In the manuscript titled IGSF11-mediated immune modulation: Unlocking a novel pathway in emerging cancer immunotherapies.The overall structure and expression of the manuscript are well-organized. However, optimizations on the framework and some detailed expressions are recommended. Specific revision suggestions are as follows:
1.In Section 2 "Structural and Functional Overview of IGSF11":
- On the basis of the existing descriptions of gene and protein structures, the classification of IGSF11 within the immunoglobulin superfamily (IGSF) should be clarified, with a brief analysis of its family attributes.
- For the part of known physiological roles, if the content is limited, it is advisable to elaborate with specific experimental evidence (e.g., phenotypes of gene knockout mice, results of in vitro cell experiments). If the known functions are scarce, reasonable inferences can be made from the perspective of "expression-function" correlation.
2.Section 7 should focus more specifically on "unsolved issues".
We hope these suggestions will be helpful for improving the manuscript. Comments on the Quality of English Language The English expression of the article features accurate terminology and a clear structure. However, it is advisable to conduct checks and revisions regarding grammatical consistency and precision,for example:"demonstrate synergistic effects" → "demonstrates synergistic effects"Author Response
In the manuscript titled IGSF11-mediated immune modulation: Unlocking a novel pathway in emerging cancer immunotherapies. The overall structure and expression of the manuscript are well-organized. However, optimizations on the framework and some detailed expressions are recommended. Specific revision suggestions are as follows:
Comment 1.In Section 2 "Structural and Functional Overview of IGSF11":
On the basis of the existing descriptions of gene and protein structures, the classification of IGSF11 within the immunoglobulin superfamily (IGSF) should be clarified, with a brief analysis of its family attributes.
Response 1: Thank you for your insightful comment. We have now included a short paragraph in the revised manuscript (MS Word manuscript, page no. 5, section 2.1, line 179-185) clarifying the classification of IGSF11 within the immunoglobulin superfamily and its family attributes. Please find the attached paragraph below for your quick reference.
Recently, the V-Set and immunoglobulin (Ig) domain-containing (VSIG) family, which is classified under IgSF and shares structural similarities with the B7 family proteins, has gained recognition as a possible immune checkpoint that plays a role in tumor evasion. Currently, this family consists of eight members, namely VSIG1, VSIG2, VSIG3, VSIG4, VSIG8, VSIG9, VSIG10, and VSIG10 L, all of which are type I transmembrane proteins expressed by various immune and nonimmune cells, many of which exhibit immunosuppressive characteristics [51]. Among them, IGSF11, also known as BT-IgSF, BTIGSF, CT119, CXADRL1, and VSIG3 is a notable member.
Comment 2: For the part of known physiological roles, if the content is limited, it is advisable to elaborate with specific experimental evidence (e.g., phenotypes of gene knockout mice, results of in vitro cell experiments). If the known functions are scarce, reasonable inferences can be made from the perspective of "expression-function" correlation.
Response 2: Thank you for your suggestion. We have incorporated recent published work on IGSF11 knockdown and its functional analysis to strengthen the context (MS Word manuscript, Page 7, section 2.3, line 245-249). Please find the paragraph attached below.
Shekari and colleagues [20] showed that decreasing the interactions between IGSF11 and VISTA after silencing IGSF11 could change the cytokine profile of human T-cells towards a more pro-inflammatory state, which aids T-cells in becoming more activated and transforming the immunosuppressive tumor microenvironment into a more anti-tumoral one.
Furthermore, we have relocated part of the content from Section 2.1 (Gene and Protein Structure) to Section 2.3 (known physiological roles), where it aligns more appropriately with the thematic flow of the manuscript (MS Word manuscript page 7, section 2.3, line 230-236)
IGSF11 plays multiple roles in cellular processes, including cell adhesion, migration [22], proliferation, and differentiation [23]. It is also involved in the formation of synapses [24], maintaining the blood-testis barrier [25], and supporting meiotic processes in both somatic and germ cells [26]. In addition to these functions, IGSF11 acts as a ligand for the immune checkpoint protein VISTA, helping to regulate immune cell activity, particularly in T-cells (Figure 2).
Comment 3: .Section 7 should focus more specifically on "unsolved issues".
Response 3: Thank you for your comment. We would like to clarify that Section 7 already includes a focused discussion on the key unsolved issues related to IGSF11-targeted immunotherapies. This includes the limited understanding of IGSF11’s intracellular signaling mechanisms (MS Word manuscript, page 21, section 7.1, line 658-666), the need for clinical validation of emerging agents such as IMT-18 and SG7, and the absence of predictive biomarkers to guide patient selection (MS Word manuscript, section 7.2, page 21, line 674-679). We believe these address the concern and provide a clear direction for future research.
We hope these suggestions will be helpful for improving the manuscript.
Comment 4: The English expression of the article features accurate terminology and a clear structure. However, it is advisable to conduct checks and revisions regarding grammatical consistency and precision,for example:"demonstrate synergistic effects" → "demonstrates synergistic effects"
Response 4: Thank you for your observation. We appreciate your attention to linguistic precision. We have carefully reviewed the manuscript for grammatical consistency and have made necessary corrections, including the suggested revision from “demonstrate synergistic effects” to “demonstrates synergistic effects,” (MS Word manuscript, page 22, section 8, line 694), along with other similar adjustments throughout the text to enhance clarity and accuracy (MS Word manuscript,line16,19,20,35,42,43,50 and so on).
Reviewer 2 Report
Comments and Suggestions for Authors
The review presents the topic in an orderly way, including sections providing an overview of immune checkpoints in cancer, the emergence of novel immune checkpoint ligands, an overview of IGSF11 in cancers, and what is known about IGSF11-mediated immune modulation. This is followed by comments on IGSF11 targeting as a therapeutic approach and future perspectives.
Several sections of the text need additional documentation. See, for example, Lines 52-67; Lines 104-122; Lines 169-182. Further on, whole paragraphs and entire subsections summarize a single report. These could be combined into shorter summaries. Some of the details are less important than the conclusions and meaning in the context of the larger story.
Lines 192-201 – The term IGSF11 has already been used (line 138, 154, 156, 160, 163, and so forth), why keep spelling it out?
Line 229-240 - “Studies have shown that IGSF11 gene expression is markedly elevated in patients with high-grade gliomas.” Only one study is cited. Why not begin the paragraph Ghouzlani and colleagues (21) have shown that….?
Line 203, 250 - Watanabe et al.? Watanabe and colleagues? Are both from reference 18? If so, please indicate.
Line 275-284 – If the entire paragraph is to summarize a single paper, it would be useful to indicate at the beginning. Shakari et al reported (20) reported that…..
Line 290 – Olbromski et al., (19) showed….
Line 298 – spelling - Reference
Lines 300-344 – Is this analysis original? Please indicate. Reference for UALCAN? Source of data in Figure 3? If this is original, some description of the process is in order.
Table 3 separated from its title?
Line 400 – incomplete sentence - Please review for completeness of sentences, accuracy of subject in the sentences. For example, SG7: It is a cross-reactive… Sentences should specifically refer to the subject, here SG7, not “It”.
The writing could be more efficient by combining section 5.1 into a brief discussion of the use of blocking antibodies and small molecules, and summarize individual antibodies and molecules in Table 3 with references? The reader is assisted by simplifying the topic where possible.
Author Response
The review presents the topic in an orderly way, including sections providing an overview of immune checkpoints in cancer, the emergence of novel immune checkpoint ligands, an overview of IGSF11 in cancers, and what is known about IGSF11-mediated immune modulation. This is followed by comments on IGSF11 targeting as a therapeutic approach and future perspectives.
Comment 1: Several sections of the text need additional documentation. See, for example, Lines 52-67; Lines 104-122; Lines 169-182. Further on, whole paragraphs and entire subsections summarize a single report. These could be combined into shorter summaries. Some of the details are less important than the conclusions and meaning in the context of the larger story.
Response 1: Thank you for your thoughtful feedback. We have reviewed the indicated sections and we have condensed paragraphs that summarize individual studies to focus on key findings and their relevance to the broader context, improving clarity and efficiency (MS Word manuscript, page 2, section 1.1, line 69-77). The paragraph is attached below.
Immunotherapy activates immune cells to target tumor cells, with immune checkpoint inhibitors (ICIs) being a major breakthrough. These inhibitors prevent immune tolerance by blocking regulatory pathways such as CTLA-4 and PD-1. The foundation of ICIs began in the early 1980s with James Allison’s discovery of tumor antigens and the T-cell receptor, followed by the identification of CTLA-4 in 1987. Although its function remained unclear for years, Allison’s 1995 research revealed CTLA-4 as a negative regulator of T-cell activation. This led to the development of ipilimumab, the first CTLA-4 blocking antibody, approved by the FDA in 2011 for metastatic melanoma [3].
(MS Word manuscript, page 3, section 1.2, line 115-123). The paragraph is attached below.
Cancer immunotherapy has become a cornerstone of modern oncology, offering durable responses through approaches such as immune checkpoint blockade (ICB), adoptive T-cell therapies, and cancer vaccines. While ICB therapies targeting PD-1, PD-L1, and CTLA-4 have revolutionized treatment, their benefits are limited to a subset of patients. Primary resistance remains a major challenge, particularly with PD-1/PD-L1 inhibitors. Although PD-L1 expression is widely used as a predictive biomarker, clinical data show that over 50% of patients with high PD-L1 levels still fail to respond, underscoring the complexity of immune resistance mechanisms beyond PD-L1 expression [9].
Comment 2: Lines 192-201 – The term IGSF11 has already been used (line 138, 154, 156, 160, 163, and so forth), why keep spelling it out?
Response 2: Thank you for your observation. The terminology has now been revised to be spelled out only at first mention, with consistent abbreviation used thereafter. (MS Word manuscript, line 163,223,229,238)
Comment 3: Line 229-240 - “Studies have shown that IGSF11 gene expression is markedly elevated in patients with high-grade gliomas.” Only one study is cited. Why not begin the paragraph, Ghouzlani and colleagues (21) have shown that….?
Response 3: Thank you for the helpful suggestion. We have revised the sentence to begin with “Ghouzlani and colleagues (21) have shown that…” to accurately reflect the single study cited (MS Word manuscript, section 3.1.1, page 9, line 274)
Comment 4: Line 203, 250 - Watanabe et al.? Watanabe and colleagues? Are both from reference 18? If so, please indicate.
Response 4: Thank you for your comment. Both mentions now use “Watanabe and colleagues,” and refer to reference 18. We’ve updated the manuscript for consistency and clarity. (section 2.3, page 7, line 242) and (MS Word manuscript, section 3.1.2, page 9, line 296)
Comment 5 Line 275-284 – If the entire paragraph is to summarize a single paper, it would be useful to indicate at the beginning. Shakari et al. (20) reported that…..
Response 5 : Thank you for the suggestion. We have revised the paragraph to begin with “Shekari et al. (20) reported that…” to indicate that the entire paragraph summarizes findings from a single study. (MS Word manuscript, page 10, section 3.1.3, line 321)
Comment 6: Line 290 – Olbromski et al., (19) showed….
Response 6: Thank you for your comment. We have revised the paragraph to begin with “Olbromski et al. (19) showed…” to clearly indicate that the findings discussed are from a single study. (MS Word manuscript, page 10, section 3.1.4, line 336)
comment 7: Line 298 – spelling – Reference
Response 7: Thank you for pointing that out. The spelling of “Reference” has been corrected in the revised manuscript. (page 10, table 1, line 345)
Comment 8: Lines 300-344 – Is this analysis original? Please indicate. Reference for UALCAN? Source of data in Figure 3? If this is original, some description of the process is in order.
Response 8: Thank you for your comment. The analysis presented in Table 2, page 13, was conducted by us through interpretation of high and low gene expression levels, compared with their normal counterparts, using graphical data obtained from publicly available UALCAN datasets, including TCGA (https://ualcan.path.uab.edu/). Added the appropriate reference to UALCAN (MS Word manuscript, page 11, section 3.1.5 line 348, reference 52). The process of analysis is briefly written (MS Word manuscript, page 11, section 3.1.5, line 348-353)
Reference 52- Chandrashekar DS, Bashel B, Balasubramanya SAH, Creighton CJ, Ponce-Rodriguez I, Chakravarthi BVSK, Varambally S. UALCAN: A Portal for Facilitating Tumor Subgroup Gene Expression and Survival Analyses. Neoplasia. 2017 Aug;19(8):649-658. doi: 10.1016/j.neo.2017.05.002.
Comment 9: Table 3 separated from its title?
Response 9: We have corrected the formatting so that Table 3 is now properly aligned with its title in the revised manuscript (MS Word manuscript, page 16, section 5, line 522)
Comment 10: Line 400 – incomplete sentence - Please review for completeness of sentences, accuracy of subject in the sentences. For example, SG7: It is a cross-reactive… Sentences should specifically refer to the subject, here SG7, not “It”.
Response 10: Thank you for your helpful observation. We have reviewed the manuscript for sentence completeness and clarity, ensuring that subjects are explicitly stated. For example, “It is a cross-reactive…” has been revised to “SG7 is a cross-reactive…” to improve precision and readability (MS Word manuscript, page 15, section 5.1.3, line 490). HMBD-002 is a first in class… is also revised (MS Word manuscript, page 15, section 5.1.3, line 469)
Comment 11:The writing could be more efficient by combining section 5.1 into a brief discussion of the use of blocking antibodies and small molecules, and summarize individual antibodies and molecules in Table 3 with references? The reader is assisted by simplifying the topic where possible.
Response 11: Thank you for your constructive suggestion. We have revised Section 5.1 to provide a more concise and integrated discussion of blocking antibodies and small molecule inhibitors targeting the IGSF11–VISTA axis. (MS Word manuscript, section 5.1, page 15, line 453-459). For your quick reference, we have added the paragraph here as well.
Monoclonal antibodies such as IMT-18 disrupt the IGSF11–VISTA interaction, restoring T-cell function within the tumor microenvironment. HMBD-002 binds to a critical domain of VISTA involved in IGSF11 engagement, while SG7 directly blocks the VISTA–IGSF11 interface. Additionally, small molecule inhibitors like K284-3046 and sinefungin are being explored to interfere with IGSF11’s signaling pathways, further enhancing immune activation against tumors.
Individual agents, including IMT-18, HMBD-002, SG7, K284-3046, and sinefungin, are summarized in Table 3 along with relevant references. This restructuring aims to improve readability and streamline the presentation of therapeutic strategies.